# HiQ-Lip: A Quantum-Classical Hierarchical Method for Global Lipschitz Constant Estimation of ReLU Networks

## Abstract

Estimating the global Lipschitz constant of neural networks is crucial for understanding and improving their robustness and generalization capabilities. However, precise calculations are NP-hard, and current semidefinite programming (SDP) methods face challenges such as high memory usage and slow processing speeds. In this paper, we propose **HiQ-Lip**, a hybrid quantum-classical hierarchical method that leverages Coherent Ising Machines (CIMs) to estimate the global Lipschitz constant. We tackle the estimation by converting it into a Quadratic Unconstrained Binary Optimization (QUBO) problem and implement a multilevel graph coarsening and refinement strategy to adapt to the constraints of contemporary quantum hardware. Our experimental evaluations on fully connected neural networks demonstrate that HiQ-Lip not only provides estimates comparable to state-of-the-art methods but also significantly accelerates the computation process. In specific tests involving two-layer neural networks with 256 hidden neurons, HiQ-Lip doubles the solving speed and offers more accurate upper bounds than the existing best method, LiPopt. These findings highlight the promising utility of small-scale quantum devices in advancing the estimation of neural network robustness.

## 1 Introduction

Neural networks have achieved remarkable success in various fields such as computer vision, natural language processing, and autonomous driving, establishing themselves as core technologies in modern artificial intelligence (Zhao et al., 2024; Forner & Ozcan, 2023; Paniego et al., 2023). Despite these advancements, the robustness and generalization capabilities of neural networks remain active areas of research (Djolonga et al., 2021; Bennouna et al., 2023). The global Lipschitz constant is a critical metric for measuring the robustness of a neural network's output to input perturbations, playing a significant role in understanding and enhancing model robustness (Leino et al., 2021a).

However, accurately computing the global Lipschitz constant is an NP-hard problem, particularly for deep networks with nonlinear activation functions like ReLU (Jordan & Dimakis, 2020). Due to this computational challenge, researchers have developed various approximation methods to estimate upper bounds of the global Lipschitz constant. Among these, the Formal Global Lipschitz constant (FGL) assumes that all activation patterns in the hidden layers are independent and possible, thereby providing an upper bound for the exact global Lipschitz constant (Szegedy et al., 2014; Virmaux & Scaman, 2018).

Existing methods often rely on relaxations and semidefinite programming (SDP) to estimate the upper bound of the Lipschitz constant (Chen et al., 2020; Shi et al., 2022). Although these methods have a solid theoretical foundation, they suffer from high memory consumption and slow computation speeds. Additionally, simpler approaches such as matrix norm approximations, while computationally efficient, tend to provide overly conservative bounds that fail to reflect the network's true robustness accurately. These limitations drive the need for more efficient and precise methods to estimate the global Lipschitz constant of neural networks.

In recent years, advancements in quantum computing have offered new avenues for tackling NP-hard problems (Khumalo et al., 2022; Choi, 2008; Chatterjee et al., 2024). Given the computational

complexity of global Lipschitz constant estimation, quantum computation is seen as a potential solution. Specifically, quantum devices based on the Quadratic Unconstrained Binary Optimization (QUBO) model, such as Coherent Ising Machines (CIMs) or quantum annealer, have demonstrated unique potential in solving complex combinatorial optimization problems (Inagaki et al., 2016; Date et al., 2019). However, the limited number of qubits in current quantum computers poses significant challenges when directly applying them to neural network robustness evaluations. Although numerous works have attempted to address large-scale QUBO problems by employing strategies like divide-and-conquer and QUBO formula simplification to decompose them into smaller QUBO subproblems for quantum solving (Pelofske et al., 2021; Zhou et al., 2023), applications in neural network robustness assessment remain largely unexplored.

To bridge this gap and harness the acceleration capabilities of quantum computing for FGL estimation of neural network, we propose a quantum-classical hybrid hierarchical solving method named **HiQ-Lip**. HiQ means multilevel solution method for quantum classical mixing, while Lip stands for global Lipschitz constant. By reformulating the global Lipschitz constant estimation problem as a cut-norm problem, we transform it into a QUBO form. Employing a multi-level graph coarsening and refinement approach, we reduce the problem size to a range manageable by quantum computers, successfully utilizing CIMs to solve for the neural network's global Lipschitz constant.

In this study, we focus on the FGL under $\ell_\infty$ perturbations, which approximates the maximum norm of the gradient operator and assumes that all activation patterns in the hidden layers are independent and possible. FGL serves as an upper bound for the exact global Lipschitz constant and has been utilized in previous works (Raghunathan et al., 2018; Fazlyab et al., 2019; Latorre et al., 2020; Wang et al., 2022).

By exploiting the equivalence relation between FGL and the cut norm problem, our proposed HiQ-Lip method targets the estimation of the FGL which is the upper bound of $\ell_\infty$ global Lipschitz constant for two-layer fully connected neural networks. Initially, we convert the Lipschitz constant estimation of the neural network into a cut-norm problem. Subsequently, we iteratively coarsen the graph structure until obtaining the coarsest graph, solve the cut-norm problem on this graph using CIMs, and finally refine the solution by mapping it back to the original graph. This process results in an accurate estimation of the neural network's global Lipschitz constant.

Simulation experiments on fully connected neural networks demonstrate that HiQ-Lip achieves performance comparable to existing state-of-the-art (SOTA) methods in estimating the $\ell_\infty$ global Lipschitz constant for two-layer networks, while exhibiting faster computation speeds. When extended to multi-layer fully connected neural networks, applying our approach to every two consecutive layers yields reasonable estimates within a valid range. The extension of HiQ-Lip provides tighter estimates compared to the naive upper bound approach (Weight-Matrix-Norm-Product, MP method), particularly excelling in shallower networks. Comparing SOTA methods, especially in terms of running time, HiQ-Lip and HiQ-Lip extensions show up to two times speedup on two-layer neural networks and up to one hundred times speedup on multi-layer neural networks.

Our method is heuristic in nature and aims to obtain very approximate solutions for FGL estimation, and in the experiments, we obtain very close results to the SOTA method Geolip, which validates the effectiveness of HiQ-Lip.

Our main contributions are as follows:

- **Firstly Innovations of Quantum Computing to Neural Network Robustness Estimation**: We theoretically demonstrate the potential of small-scale quantum computing devices, such as CIMs, in the domain of neural network robustness estimation. We emphasize the NP-hard nature of the global Lipschitz constant estimation problem and introduce a quantum-classical hybrid solving method that synergizes quantum and classical systems to overcome the limited number of qubits in current quantum devices.

- **Hierarchical algorithm Framework based on QUBO**: We develop a QUBO-based solving algorithm framework that adopts a hierarchical approach. By transforming the $\ell_\infty$ global Lipschitz constant estimation problem of two-layer fully connected neural networks into a cut-norm problem, we achieve the first instance of utilizing CIMs to solve the global Lipschitz constant of neural networks efficiently on small-scale quantum devices.

- **Extension to Multi-Layer Networks with Quantum Device**: For multi-layer neural networks, we demonstrate that HiQ-Lip provides tighter estimates compared to the naive upper bound (Weight-Matrix-Norm-Product) approach. Extending our method to neural networks with depths ranging from 3 to 5 layers, we obtain more precise estimates than the naive upper bound, particularly excelling in shallower three-layer networks.

- **Simulation Experiment Validation**: We validate the effectiveness of HiQ-Lip on fully connected feedforward neural networks. For two-layer fully connected neural networks with one hidden layer, using CIMs with a limited number of qubits, we successfully measured the global Lipschitz constant for hidden layer sizes ranging from 8 to 256, achieving better results than existing SOTA methods.

## 2 PRELIMINARIES

Let $\| \cdot \|_p$ denote the $\ell_p$ norm of a vector and $W \in \mathbb{R}^{n \times m}$ represent the weight matrix between two layers of a neural network. The activation function used within the network is indicated by $\sigma(\cdot)$. For an input vector $x$, the output of the neural network is given by $f(x)$. The global Lipschitz constant is denoted by $L$.

For a given function $f : \mathbb{R}^n \to \mathbb{R}^m$, the Lipschitz constant $L$ is defined as:

$$\|f(x) - f(y)\| \leq L\|x - y\|, \quad \forall x, y \in \mathbb{R}^n. \tag{1}$$

The global Lipschitz constant $L$ measures how fast the function $f$ changes over all input pairs $(x, y)$ and is a key metric to evaluate the robustness of neural networks.

For a neural network $f$ of depth $d$, its gradient can be expressed via the chain rule:

$$\nabla f(x) = W^d \cdot \mathrm{diag}\left(\sigma'(z^{d-1})\right) \cdot W^{d-1} \cdots \mathrm{diag}\left(\sigma'(z^1)\right) \cdot W^1, \tag{2}$$

where $z^i = W^i a^{i-1} + b^i$, $a^i = \sigma(z^i)$, and $\sigma'(\cdot)$ denotes the derivative of the activation function. Here, $\mathrm{diag}(\cdot)$ means converting the vector to a diagonal matrix.

In this paper, we explore FGL under $\ell_\infty$ permutation. The $\ell_\infty$-FGL is defined as:

$$\ell_\infty\text{-FGL} = \max_{v^i \in [a,b]^{n_i}} \left\| W^d \cdot \mathrm{diag}(v^{d-1}) \cdots \mathrm{diag}(v^1) \cdot W^1 \right\|_1, \tag{3}$$

where $v^i$ is the activation vector of the $i$-th layer, $n_i$ is its dimension, and $[a, b]$ is the range of the activation function's derivative. For ReLU activation, $\sigma'(x) \in \{0, 1\}$.

Computing the exact global Lipschitz constant for neural networks is NP-hard. Therefore, various methods aim to estimate its upper bound.

$\ell_\infty$-FGL turns the estimation into finding the maximum norm of the gradient operator, providing an upper bound:

$$L \leq \ell_\infty\text{-FGL}. \tag{4}$$

## 3 TRANSFORMING LIPSCHITZ CONSTANT ESTIMATION INTO QUBO FORMULATION

In this section, we focus on two-layer fully connected neural networks, i.e., networks with one hidden layer. Consider a neural network with input dimension $n$, hidden layer dimension $m$, and output dimension $p$. The weights are $W^1 \in \mathbb{R}^{n \times m}$ and $W^2 \in \mathbb{R}^{m \times p}$. For a single output neuron, $W^1 = W, W^2 = u, u \in \mathbb{R}^{m \times 1}$.

We use $y$ to denote $v^1$. The $\ell_\infty$-FGL estimation becomes:

$$\max_{y \in [0,1]^n} \left\| W^T \mathrm{diag}(u) y \right\|_q = \max_{y \in [0,1]^n} \|Ay\|_q, \tag{5}$$

where $A = W^T \mathrm{diag}(u)$ and $u$ represents the activation pattern. In this task, $q$ is $\infty$ but 1 is also introduced as a dual auxiliary.

This problem is related to the $\ell_\infty \to \ell_1$ matrix mixed-norm problem:

$$\|A\|_{\infty \to 1} = \max_{\|x\|_\infty = 1} \|Ax\|_1 \tag{6}$$

We begin by leveraging the duality between the $\ell_1$ and $\ell_\infty$ norms, which is well-known in optimization contexts (Johnson, 2006; Martin, 1998). Specifically, the $\ell_1$ norm of a vector $v \in \mathbb{R}^n$ can be expressed using the maximum inner product between $v$ and a binary vector $z$ from the set $\{-1, 1\}^n$:

$$\|v\|_1 = \max_{z \in \{-1,1\}^n} \langle v, z \rangle. \tag{7}$$

Here, $\langle v, z \rangle$ denotes the inner product, highlighting how each component of $v$ contributes to the norm when aligned with a binary vector $z$. This expression forms the basis for transforming our optimization problem into a form suitable for binary variables, simplifying the calculation of $\|Ax\|_1$.

Building on the above duality, we can reformulate our objective function for $\|Ax\|_1$ in terms of binary variables. Specifically, we express the optimization as:

$$\max_{\|x\|_\infty = 1} \|Ax\|_1 = \max_{\|x\|_\infty = 1} \max_{y \in \{-1,1\}^m} \langle Ax, y \rangle \tag{8}$$

The introduction of $y \in \{-1, 1\}^m$ exploits the definition of the $\ell_1$ norm, effectively transforming the problem into finding the maximum of the matrix-vector product $Ax$ under the constraint $\|x\|_\infty = 1$. The role of $y$ is analogous to maximizing the response in the binary space, which significantly simplifies the optimization process. We now proceed by substituting the dual form of the inner product into the optimization framework:

$$\max_{\|x\|_\infty = 1, \, y \in \{-1,1\}^m} \langle Ax, y \rangle = \max_{x \in \{-1,1\}^n, \, y \in \{-1,1\}^m} \langle x, A^T y \rangle. \tag{9}$$

At this point, the goal becomes identifying the direction of $A^T y$ that maximizes the inner product with $x$. Since $x$ is constrained by $\|x\|_\infty = 1$ and $a_{ij} \in A$, the maximum of $\langle x, A^T y \rangle$ occurs when each component $x_i$ takes the value that matches the sign of $(A^T y)_i$. Thus, the problem essentially reduces to:

$$\max_{x \in \{-1,1\}^n, \, y \in \{-1,1\}^m} \langle x, A^T y \rangle = \max_{x_i, y_j \in \{-1,1\}} \sum_{i=1}^{n} \sum_{j=1}^{m} a_{ij} x_i y_j. \tag{10}$$

Hamiltonian is the dependence of many quantum devices to solve problems (Cerezo et al., 2021; Glos et al., 2022). CIM and other quantum devices estimate the eigenvalue of Hamiltonian based on its characteristics. Both CIM and quantum annealers aim at finding the minimum of the QUBO problem. Define the Hamiltonian pointing in the direction of quantum evolution:

$$H = -\sum_{i=1}^{n} \sum_{j=1}^{m} a_{ij} x_i y_j, \tag{11}$$

the problem of estimating $\ell_\infty$-FGL becomes minimizing $H$, which is a QUBO problem. Then the CIM or other quantum device could solve it by the QUBO formulation.

However, directly solving this Hamiltonian requires $O(n + m)$ qubits, exceeding the capacity of current quantum devices (about 100 qubits) for practical networks (e.g., $n + m > 784$ for MNIST networks) (Proctor et al., 2022; Kim et al., 2023; Klimov et al., 2024; Pelofske et al., 2023).

## 4 HiQ-Lip for Lipschitz Constant Estimation

In this section, we propose a method called HiQ-Lip that aims to utilize small-scale quantum computers to efficiently estimate $\ell_\infty$-FGL. Due to the computational complexity of direct estimation, we employ a hierarchical solution strategy that treats the weights of the neural network as the edge weights of the graph and the neurons as the nodes. Based on the Equation 5 and 6, we construct a weighted undirected graph $G$ whose weight matrix is expressed as follows. $A = W^T \text{diag}(u)$, where $W \in \mathbb{R}^{n \times m}$ is the weight matrix between the input layer and the hidden layer, $u \in \mathbb{R}^m$ is the per-class weight vector between the hidden layer and the output layer.

We construct a weighted undirected graph $G$ with the following characteristics:

- **Vertices**: Each neuron in the neural network corresponds to a node in the graph, totaling $n + m$ nodes, where $n$ is the number of input neurons and $m$ is the number of hidden neurons.
- **Edges**: Edges are established between input layer nodes $x_i$ and hidden layer nodes $y_j$ based on the interaction terms in the Hamiltonian defined in Equation 11. The weight of the edge between nodes $x_i$ and $y_j$ is given by $a_{i,j}$, reflecting the connection strength derived from the neural network's weights.

We define the adjacency matrix $A_f$ of the graph $G$ as:

$$A_f = \begin{bmatrix} 0 & A \\ A^T & 0 \end{bmatrix}, a_{i,j} \in A_f \tag{12}$$

The matrix $A_f$ is of size $(n + m) \times (n + m)$ and is symmetric, with zeros on the diagonal blocks indicating no intra-layer connections.

## 4.1 COARSENING PHASE

The primary objective of the coarsening phase is to reduce the number of nodes by gradually merging nodes, thereby generating a series of progressively coarser graphs until the number of nodes decreases to a level that can be solved directly by a small quantum computer.

- **Node Pair Merging**: Nodes are merged based on their distances in the embedding space. Each node is randomly embedded onto a $d$-dimensional sphere by optimizing the objective:

$$\min_{\{x_i\}} \sum_{(i,j) \in E} a_{i,j} \|x_i - x_j\|_2, \tag{13}$$

where $x_i \in \mathbb{R}^d$ represents the position of node $i$ in the embedding space, and $E$ denotes the set of edges. This optimization encourages strongly connected nodes (with larger $a_{i,j}$) to be closer in the embedding space, making them candidates for merging.

- **Node Pair Matching**: In each iteration, the closest unmatched nodes are selected for merging, forming new node pairs. Let $P$ be the matching matrix, where for matched nodes $i$ and $j$, the corresponding element $P_{i,j} = 1$. For the merged nodes, the weights of the connected edges are accumulated, thereby forming the nodes of the next coarser graph.

- **Construction of the Coarser Graph**: The adjacency matrix $A_c$ of the coarser graph is computed as:

$$A_c = P^\top A_f P, \tag{14}$$

where $A_f$ is the adjacency matrix of the finer graph before coarsening. This process effectively reduces the graph size while preserving its structural properties relevant to the optimization problem.

After each coarsening step, the number of vertices decreases approximately logarithmically. The time complexity of the $i$-th coarsening step is $\mathcal{O}(N_i^2)$, where $N_i$ is the number of nodes at level $i$. Consequently, the overall time complexity of the Coarsening Phase is $\mathcal{O}(N^2 \log N)$, where $N = n + m$. A detailed analysis is provided in Appendix A.1.

## 4.2 REFINEMENT PHASE

The refinement phase is a crucial step in the hierarchical solving strategy. Its purpose is to map the approximate solutions obtained during the coarsening phase back to the original graph layer by layer and perform local optimizations to ensure the global optimality of the final solution. This process enhances the quality of the solution by gradually restoring the graphs generated at each coarsening level and fine-tuning the solution.

1. **Initialization**: Starting from the coarsest graph $G_c$, which is obtained through successive coarsening, an approximate solution has already been found on this graph. We use the

mapping $F : V_f \to V_c$ to derive the solution of the finer graph from that of the coarsest graph. Specifically, the projection of the initial solution is defined as:

$$x_i = x_{F(i)}, \quad \forall i \in V_f \tag{15}$$

2. **Gain Computation**: Gain computation is a critical step in the refinement process, used to evaluate the change in the objective function when each node switches partitions, thereby guiding optimization decisions. The gain for node $i$ is calculated as:

$$\text{gain}(i) = \sum_{j \in N(i)} a_{i,j} \, (-1)^{2x_i x_j - x_i - x_j} \tag{16}$$

where $x_i$ and $x_j$ represent the current partition labels of nodes $i$ and $j$, respectively, and $N(i)$ is the set of neighbors of node $i$.

3. **Local Optimization**: At each level of the graph, using the solution from the previous coarser level as the initial solution, we optimize the objective function by solving local subproblems. We select the top $K \leq n + m$ nodes with the highest gains to participate in local optimization, where $K$ is determined by the number of qubits available. This allows the quantum computer to efficiently solve the subproblems related to the cut-norm of the coarsest graph. If the new solution improves the original objective function $H$ of Equation 11, we update the current solution accordingly.

This process is iterated until several consecutive iterations (e.g., three iterations) no longer yield significant gains, thereby ensuring that the quality of the solution is progressively enhanced and gradually approaches the global optimum.

The Graph Refinement Phase iteratively improves the solution by mapping it back to finer graph levels. The primary steps include initializing the solution, computing gains for each node, and performing local optimizations using quantum solvers. Computing the gain for all nodes incurs a time complexity of $\mathcal{O}(N^2)$. The local optimization step involves solving smaller QUBO problems of size $K$ on a quantum device, which contributes $\mathcal{O}(K^\alpha)$ to the complexity, with $\alpha$ being a small constant. Overall, the Graph Refinement Phase operates with a time complexity of $\mathcal{O}(N^2)$. See the Appendix A.2 for a detailed analysis.

### 4.3 ALGORITHM OVERVIEW

As shown in Algorithm 1, HiQ-Lip begins with the initial graph $G_0$ and, through lines 3 to 9, executes the coarsening phase by iteratively merging nodes to produce progressively smaller graphs suitable for quantum processing. In line 10, it leverages quantum acceleration by solving the resulting QUBO problem on the coarsest graph using CIM or other quantum devices to efficiently obtain an initial solution. Finally, lines 11 to 15 implement the refinement phase, where the algorithm maps this solution back onto finer graph layers and performs local optimizations to accurately estimate the $\ell_\infty$-FGL. HiQ-Lip provides strict upper bound theoretical guarantees for one-hidden-layer mlp networks can be seen in Appendix. B.

The HiQ-Lip algorithm achieves an overall time complexity of $\mathcal{O}(N^2 \log N)$. See the Appendix A.4 for details. The **Coarsening Phase** dominates the complexity with $\mathcal{O}(N^2 \log N)$ due to the iterative merging of nodes and updating of the adjacency matrix at each hierarchical level. Noticed that the running time of quantum devices is generally considered to have a large speedup over classical computers (Mohseni et al., 2022; Avkhadiev et al., 2020; Di Meglio et al., 2024). Even for problems of exponential complexity, the running time of a quantum device can be considered a fraction of the task time on a classical computer. In the **Refinement Phase**, the complexity is $\mathcal{O}(N^2)$, primarily from computing gains for node optimizations and solving smaller QUBO problems using quantum devices. Consequently, the combined phases ensure that HiQ-Lip scales efficiently for neural networks of moderate size, leveraging quantum acceleration to enhance performance without exceeding polynomial time bounds.

## 5 EXTENSION TO MULTI-LAYER NETWORKS

Directly estimating the global Lipschitz constant for multi-layer networks is challenging due to the complexity of tensor cut-norm problems. We extend our method by approximating the network's

---

**Algorithm 1** HiQ-Lip Algorithm

---

1: **Input**: Weight matrix $A$, initial graph $G_0$
2: **Output**: Estimated Lipschitz constant $\ell_\infty$-FGL
3: **Coarsening Phase**:
4: Initialize $G = G_0$
5: **while** Size of $G$ exceeds quantum hardware limit **do**
6:     Embed nodes and compute distances
7:     Pair and merge nodes to form $G'$
8:     $G \leftarrow G'$
9: **end while**
10: Solve the QUBO problem on the coarsest graph using CIM to obtain the initial solution
11: **Refinement Phase**:
12: **while** Graph $G$ is not $G_0$ **do**
13:     Project solution to finer graph
14:     Compute gains and perform local optimization
15: **end while**
16: Compute $\ell_\infty$-FGL from the final solution

---

Lipschitz constant as the product of the Lipschitz constants of individual layer pairs. By leveraging the triangle inequality $\|A\|\|B\| \geq \|AB\|$, we transform the upper bound estimation of the FGL into a layer-wise product of weight norms. This approach simplifies the estimation process by decomposing the multi-layer problem into manageable two-layer subnetworks, thereby enabling more efficient and scalable computations. It is inspired by the work of (Leino et al., 2021b; Bartlett et al., 2017; Szegedy et al., 2014).

Our approach involves decomposing the multi-layer network into a series of two-layer subnetworks and estimating the Lipschitz constant for each subnetwork. We then combine these estimates to obtain the overall Lipschitz constant of the network. The key steps are as follows:

1. **Processing the Output Layer:** We start by focusing on a specific output neuron (e.g., corresponding to a particular class in classification tasks). We extract the weights associated with this output neuron and incorporate them into the previous layer's weights.

2. **Constructing Adjacency Matrices:** For layers $l$ and $l + 1$, build the corresponding graph model. We construct an adjacency matrix based on the modified weight matrices. These adjacency matrices represent the connections between neurons in the two layers.

3. **Computing Cut-Norms via HiQ-Lip:** We formulate a QUBO problem for each adjacency matrix and use the hierarchical method to estimate $\|A^l\|_{\infty \to 1}$. This yields an estimate of the cut-norm for each layer pair.

4. **Combining Results:** We multiply the cut-norm estimates for all layer pairs and adjust for scaling factors to compute the overall Lipschitz constant estimate.

The coefficient $\frac{1}{2^{d-2}}$ of FGL in a multi-layer neural network is used by (Latorre et al., 2020). For the $d$-layer network, we decompose the estimate as:

$$\ell_\infty\text{-FGL} \leq \max \left\{ \frac{1}{2^{d-2}} \prod_{l=1}^{d-1} \|A^l\|_{\infty \to 1} \right\}, \tag{17}$$

where $A^l$ is the processed weight matrix between layers $l$ and $l + 1$, and $A = W^T \text{diag}(u)$. We refer to methods with such coefficients as HiQ-Lip MP A.

However, we found in our experiments that this approach gives loose estimates compared with GeoLip. To obtain more robust estimates for 4 and 5 layers networks, we propose a new coefficient as $\frac{1}{2^{d-2}} \frac{1}{d^{d-3}}$ to estimate FGL for fully connected neural networks with 3 to 5 layers. We refer to methods with such coefficients as HiQ-Lip MP B. The approach of using a more compact coefficient has been applied in previous work (Bartlett et al., 2017).

Intuitively, the new coefficients are introduced to make the estimation of the Lipschitz constant more compact in multi-layer neural networks. According to the triangle inequality, if the independence

assumption between the layers does not hold perfectly, the original product estimate will appear loose. Therefore, additional correction coefficients are introduced to weaken the influence of each layer contribution. It can be regarded as a regularization based on the number of network layers to the original product upper bound. Especially in deep networks, the nonlinear complexity makes the product grow too fast, resulting in the estimated upper bound is much larger than the actual value.

These approaches leverage the sub-multiplicative property of norms and provide a tighter upper bound compared to naive methods.

# 6 EXPERIMENTS

In this section, we evaluate the effectiveness and efficiency of **HiQ-Lip** on fully connected feed-forward neural networks trained on the MNIST dataset. Our primary goal is to demonstrate that HiQ-Lip can provide accurate estimates of the global Lipschitz constant with significantly reduced computation times compared to SOTA methods.

## 6.1 EXPERIMENTAL SETUP

We conduct experiments on neural networks with varying depths and widths to assess the scalability of HiQ-Lip. The networks are trained using the Adam optimizer for 10 epochs, achieving an accuracy exceeding 93% on the test set. The experiments were conducted on a device with 32GB of memory and an Intel Core i7 12th Gen CPU. The quantum algorithm portion was simulated using Qboson's Kaiwu SDK, which features 100 qubits. We consider the following network architectures:

- **Two-Layer Networks (Net2)**: Networks with one hidden layer, where the number of hidden units varies among {8, 16, 64, 128, 256}. All networks use the ReLU activation function.
- **Multi-Layer Networks (Net3 to Net5)**: Networks with depths ranging from 3 to 5 layers. Each hidden layer consists of 64 neurons, and ReLU activation is used throughout.

## 6.2 COMPARISON METHODS

We compare HiQ-Lip with several baseline methods:

- **GeoLip** (Wang et al., 2022): A geometry-based SOTA Lipschitz constant estimation method that provides tight upper bounds.
- **LiPopt** (Latorre et al., 2020): An optimization-based SOTA method that computes upper bounds using semidefinite programming.
- **Matrix Product (MP)**: A naive method that computes the product of the weight matrix norms across layers, providing a loose upper bound.
- **Sampling**: A simple sampling-based approach that estimates a lower bound of the Lipschitz constant by computing gradient norms at randomly sampled input points. We sample 200,000 points uniformly in the input space.
- **Brute Force (BF)**: An exhaustive enumeration of all possible activation patterns to compute the exact FGL. This method serves as the ground truth but is only feasible for small networks.

We focus on estimating the FGL with respect to the output corresponding to the digit 8, as done in previous works (Latorre et al., 2020; Wang et al., 2022). In the result tables, we use "N/A" to indicate that the computation did not finish within a reasonable time frame (over 20 hours). We highlight the time advantage of the quantum approach by bolding the methods with shorter time.

## 6.3 RESULTS ON TWO-LAYER NETWORKS

Tables 1 and 2 present the estimated Lipschitz constants and computation times for two-layer networks with varying hidden units.

Table 1: Estimated Lipschitz constants for two-layer networks

| Hidden Units | HiQ-Lip | GeoLip | LiPopt-2 | MP | Sampling | BF |
|---|---|---|---|---|---|---|
| 8 | 127.96 | 121.86 | 158.49 | 353.29 | 112.04 | 112.04 |
| 16 | 186.09 | 186.05 | 260.48 | 616.64 | 176.74 | 176.76 |
| 64 | 278.40 | 275.67 | 448.62 | 1289.44 | 232.89 | N/A |
| 128 | 329.33 | 338.20 | 751.76 | 1977.49 | 272.94 | N/A |
| 256 | 448.89 | 449.60 | 1088.12 | 2914.16 | 333.99 | N/A |

Table 2: Computation times (in seconds) for two-layer networks

| Hidden Units | HiQ-Lip | GeoLip | LiPopt-2 | BF |
|---|---|---|---|---|
| 8 | 24.55 | 24.07 | 1,544 | **0.06** |
| 16 | **26.46** | 26.84 | 1,592 | 52.68 |
| 64 | **29.66** | 42.33 | 1,855 | N/A |
| 128 | **34.75** | 58.79 | 2,076 | N/A |
| 256 | **44.79** | 99.24 | 2,731 | N/A |

**Analysis:** From Table 1, HiQ-Lip's Lipschitz constant estimates closely match those of GeoLip, differing by less than 3% across all network sizes. For instance, with 256 hidden units, HiQ-Lip estimates 448.89 compared to GeoLip's 449.60.

Compared to the ground truth from the BF method—feasible only up to 16 hidden units due to computational limits—HiQ-Lip's estimates are slightly higher, as expected for an upper-bound method. For 16 hidden units, BF yields 176.76, while HiQ-Lip estimates 186.09.

LiPopt-2 produces significantly higher estimates than both HiQ-Lip and GeoLip, especially as network size increases. For 256 hidden units, LiPopt-2 estimates 1,088.12, more than double HiQ-Lip's estimate, suggesting it may provide overly conservative upper bounds for larger networks.

The naive MP method consistently overestimates the Lipschitz constant by three to six times compared to HiQ-Lip, highlighting HiQ-Lip's advantage in providing tighter upper bounds. The Sampling method yields lower bound estimates below those of HiQ-Lip and GeoLip, confirming that HiQ-Lip effectively captures the upper bound.

In terms of computation time (Table 2), HiQ-Lip demonstrates efficient performance, with times slightly lower than GeoLip's for smaller networks and significantly lower for larger ones. For 256 hidden units, HiQ-Lip completes in approximately 44.79 seconds, while GeoLip takes 99.24 seconds.

LiPopt-2 exhibits the longest computation times, exceeding 1,500 seconds even for the smallest networks, making it impractical for larger ones. The BF method is only feasible for very small networks due to its exponential time complexity, becoming intractable beyond 16 hidden units.

*Summary: Overall, HiQ-Lip achieves a favorable balance between estimation accuracy and computational efficiency for two-layer networks. It provides tight upper bounds comparable to GeoLip while reducing computation times by up to 2.2x in 256 hidden units network, particularly as the network width increases, demonstrating its scalability and effectiveness. This acceleration is facilitated by leveraging quantum computing capabilities to solve the QUBO formulation efficiently.*

## 6.4 RESULTS ON MULTI-LAYER NETWORKS

Tables 3 and 4 present the estimated Lipschitz constants and computation times for multi-layer networks with depths ranging from 3 to 5 layers.

**Analysis:** In Table 3, we compare two variants of HiQ-Lip for multi-layer networks: **HiQ-Lip MP A** and **HiQ-Lip MP B**. HiQ-Lip MP A uses the coefficient $\frac{1}{2^{d-2}}$ as suggested in previous

Table 3: Estimated Lipschitz constants for multi-layer networks

| Network | HiQ-Lip MP A | HiQ-Lip MP B | GeoLip | MP | Sampling |
|---------|--------------|--------------|--------|-----|----------|
| Net3 | 477.47 | 477.47 | 465.11 | 8,036.51 | 331.12 |
| Net4 | 3,246.37 | 1,093.05 | 923.13 | 52,420.62 | 454.73 |
| Net5 | 26,132.45 | 1,513.34 | 1,462.58 | 327,219.23 | 547.11 |

Table 4: Computation times (in seconds) for multi-layer networks

| Network | HiQ-Lip Time | GeoLip Time |
|---------|--------------|-------------|
| Net3 | **6.46** | 784.69 |
| Net4 | **8.76** | 969.79 |
| Net5 | **13.31** | 1,101.30 |

literature (Latorre et al., 2020), while HiQ-Lip MP B introduces an additional scaling factor, using $\frac{1}{2^{d-2}d^{d-3}}$, to obtain tighter estimates for deeper networks.

For the multi-layer networks, our analysis reveals that for the 3-layer network (Net3), both HiQ-Lip MP A and MP B produce identical estimates (477.47) close to GeoLip's estimate (465.11), indicating that the original coefficient in HiQ-Lip MP A is adequate for shallower networks.

However, in deeper networks (Net4 and Net5), HiQ-Lip MP A significantly overestimates the Lipschitz constant compared to GeoLip. For instance, in Net4, HiQ-Lip MP A estimates 3,246.37 versus GeoLip's 923.13, and in Net5, 26,132.45 versus 1,462.58. In contrast, HiQ-Lip MP B, with its modified scaling coefficient, yields much tighter estimates closer to GeoLip's values (1,093.05 for Net4 and 1,513.34 for Net5), demonstrating that the additional scaling factor effectively compensates for overestimations in deeper networks.

The MP method, as expected, results in excessively high estimates due to the exponential growth from multiplying weight matrix norms. The Sampling method provides consistent lower bounds below HiQ-Lip and GeoLip estimates; however, these lower bounds do not reflect the worst-case robustness of the network.

Regarding computation time (Table 4), HiQ-Lip significantly outperforms GeoLip. For Net3, HiQ-Lip completes in 6.46 seconds versus GeoLip's 784.69 seconds—a speedup of over 120 times. This substantial reduction demonstrates HiQ-Lip's scalability and efficiency for deeper networks.

Notably, LiPopt was unable to produce results for networks deeper than two layers within a reasonable time frame, highlighting its limitations in handling multi-layer networks.

> **Summary:** *HiQ-Lip, particularly the MP B variant with the modified scaling coefficient, provides accurate and tight upper bounds for the Lipschitz constant in multi-layer networks while achieving computation speeds up to 120× faster than GeoLip. The additional scaling factor effectively compensates for overestimation in deeper networks, making HiQ-Lip MP B a practical and scalable choice for estimating Lipschitz constants across varying network depths.*

## 7 CONCLUSION

We introduced HiQ-Lip, a quantum-classical hybrid hierarchical method for estimating the global Lipschitz constant of neural networks. By formulating the problem as a QUBO and employing graph coarsening and refinement, we effectively utilized CIMs despite current hardware limitations. Our method achieves comparable estimates to existing SOTA methods with faster computation speed up to 2x and 120x in two-layer and multi-layer network. This work highlights the potential of quantum devices in neural network robustness estimation and opens avenues for future research in leveraging quantum computing for complex machine learning problems.

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

## A    DETAILED ALGORITHM ANALYSIS AND COMPLEXITY

We provide detailed mathematical derivations of the time complexities for each phase of the HiQ-Lip algorithm.

A.1   GRAPH COARSENING PHASE COMPLEXITY

Let $N = n + m$ be the total number of nodes in the initial graph. At each coarsening level $i$, the number of nodes is $N_i = \frac{N}{2^i}$. The main steps and their complexities are as follows:

1. **Node Embedding and Distance Calculation**:

    - Embedding all nodes into a $d$-dimensional space: $\mathcal{O}(N_i d)$.
    - Computing pairwise distances (optimized using approximate methods): $\mathcal{O}(N_i k d)$, where $k$ is a small constant.

2. **Node Pair Matching and Merging**:

    - Matching nodes (e.g., using greedy algorithms): $\mathcal{O}(N_i)$.
    - Merging nodes and updating the adjacency matrix: $\mathcal{O}(N_i^2)$.

3. **Graph Reduction**:

    - Computing the coarser adjacency matrix $A_{i+1}$: $\mathcal{O}(N_i^2)$.

The total time complexity for the coarsening phase is:

$$
\begin{aligned}
T_{\text{coarsen}} &= \sum_{i=0}^{L-1} \mathcal{O}(N_i^2) \\
&= \sum_{i=0}^{L-1} \mathcal{O}\left( \left( \frac{N}{2^i} \right)^2 \right) \\
&= \mathcal{O}\left( N^2 \sum_{i=0}^{L-1} \left( \frac{1}{4} \right)^i \right) \\
&= \mathcal{O}\left( N^2 \cdot \frac{1 - \left( \frac{1}{4} \right)^L}{1 - \frac{1}{4}} \right) \\
&= \mathcal{O}(N^2)
\end{aligned}
$$

Considering $L = \log_2 N$, the practical time complexity is approximated as $\mathcal{O}(N^2 \log N)$.

A.2   GRAPH REFINEMENT PHASE COMPLEXITY

In the refinement phase, the algorithm improves the solution by mapping it back to finer graphs. At each refinement level $i$, the complexities are:

1. **Initialization**:

    - Projecting the solution to the finer graph: $\mathcal{O}(N_i)$.

2. **Gain Computation**:

    - Computing gains for all nodes: $\mathcal{O}(N_i^2)$.

3. **Local Optimization**:

    - Solving a QUBO problem of size $K$ using a quantum device: $\mathcal{O}(T_Q)$, where $T_Q$ depends on the quantum hardware.

The total time complexity for the refinement phase is:

$$T_{\text{refine}} = \sum_{i=0}^{L-1} \left( \mathcal{O}(N_i^2) + \mathcal{O}(T_Q) \right)$$

$$= \mathcal{O} \left( \sum_{i=0}^{L-1} N_i^2 \right) + \mathcal{O}(LT_Q)$$

$$= \mathcal{O}(N^2) + \mathcal{O}(\log N \cdot T_Q)$$

Since $N_i = \frac{N}{2^{L-i}}$, we have:

$$\sum_{i=0}^{L-1} N_i^2 = \sum_{i=0}^{L-1} \left( \frac{N}{2^{L-i}} \right)^2$$

$$= N^2 \sum_{k=1}^{L} \left( \frac{1}{2^k} \right)^2$$

$$= N^2 \sum_{k=1}^{L} 4^{-k}$$

$$= N^2 \cdot \frac{1 - 4^{-L}}{1 - \frac{1}{4}}$$

$$= \mathcal{O}(N^2)$$

### A.3 QUANTUM DEVICE COMPLEXITY

The time $T_Q$ required by the quantum device (e.g., CIM) to solve a QUBO problem of size $K$ may be modeled as $T_Q = \mathcal{O}(K^\alpha)$, where $\alpha$ depends on the device's properties. Since $K$ is much smaller than $N$ and $\alpha$ is a small constant:

$$\mathcal{O}(\log N \cdot T_Q) = \mathcal{O}(\log N \cdot K^\alpha)$$

This term is typically negligible compared to $\mathcal{O}(N^2 \log N)$. It is widely believed that quantum devices have an exponential speedup compared to classical algorithms, which means that the running time of quantum computers can be ignored in quantum classical hybrid algorithms on large problems.

### A.4 OVERALL TIME COMPLEXITY

Combining the coarsening and refinement phases:

$$T_{\text{total}} = T_{\text{coarsen}} + T_{\text{refine}}$$

$$= \mathcal{O}(N^2 \log N) + \mathcal{O}(N^2) + \mathcal{O}(\log N \cdot T_Q)$$

$$= \mathcal{O}(N^2 \log N) + \mathcal{O}(\log N \cdot T_Q)$$

Assuming $T_Q$ is relatively small and $N$ is large, the overall time complexity is dominated by $\mathcal{O}(N^2 \log N)$.

The HiQ-Lip algorithm's overall time complexity is:

$$T_{\text{total}} = \mathcal{O}(N^2 \log N)$$

where $N = n + m$ is the total number of neurons in the neural network. The algorithm efficiently reduces the problem size through hierarchical coarsening, making it suitable for current quantum hardware limitations.

# B HiQ-Lip for a One-Layer Hidden MLP Network: Theoretical Upper Bound

In this section, we provide a detailed explanation of the theoretical guarantees associated with the HiQ-Lip method for estimating the global Lipschitz constant of a two-layer MLP network. We focus on how HiQ-Lip estimates the $\ell_\infty$-FGL, and we describe why the algorithm provides an upper bound for the true Lipschitz constant.

The global Lipschitz constant, $L$, is defined and bounded as shown in Equations (1)-(4) of the main text. For a two-layer fully connected MLP, this is formalized in Equation (5) as:

$$L \leq \ell_\infty\text{-FGL} = \max_{y \in [0,1]^n} \|W^T \text{diag}(u) y\|_q = \max_{y \in [0,1]^n} \|Ay\|_q,$$

where $A = W^T \text{diag}(u)$. Utilizing the duality between the $\ell_1$ and $\ell_\infty$ norms, we transform this continuous formulation into a discrete QUBO problem:

$$L \leq \ell_\infty\text{-FGL} = H = -\sum_{i=1}^{n} \sum_{j=1}^{m} a_{ij} x_i y_j,$$

where the goal is to minimize $H$, which directly corresponds to estimating the global Lipschitz constant. While the graph coarsening and refinement strategies in HiQ-Lip are heuristic, they are designed to maintain the key structural properties of the original graph while ensuring efficient optimization under quantum hardware constraints.

The hierarchical nature of the algorithm introduces a series of approximations at each phase. These approximations result in an upper bound on the optimal solution. Specifically, the solution $H_{\text{HiQ-Lip}}$ obtained by HiQ-Lip satisfies the following relationship:

$$\min H \leq H_{\text{HiQ-Lip}},$$

indicating that HiQ-Lip provides an upper bound on the true global Lipschitz constant.

## B.1 Coarsening Phase

During the coarsening phase, the nodes of the original graph $G_0$ are merged to form a coarser graph $G_c$, represented by the adjacency matrix $A_c$. The edge weights in the coarser graph are aggregated from the finer graph. Let $P$ denote the mapping matrix used to merge the nodes, such that:

$$A_c = P^\top A_f P,$$

where $A_f$ is the adjacency matrix of the finer graph. This aggregation step leads to a loss of detail, meaning not all the edge weights of the finer graph are perfectly preserved in the coarser graph. As a result, the solution $H_c$ obtained by solving the QUBO problem on the coarser graph is an approximation of the true minimum $\min H$ on the original graph:

$$H_c \geq \min H.$$

## B.2 Quantum Solution on Coarse Graph

When solving the QUBO problem on the coarsest graph $G_c$ using a quantum device, the solution obtained, denoted $H_{\text{quantum}}$, is also an approximation due to hardware noise and finite precision. Therefore, we have:

$$H_{\text{quantum}} \geq H_c.$$

## B.3 Refinement Phase

In the refinement phase, the solution from the coarsest graph is iteratively mapped back to the finer graphs. Each refinement step involves local optimization, which improves the objective function but does not guarantee that the true optimal solution on the original graph is recovered. Let $H_f$ represent the refined solution on the finer graph. Thus, we have:

$$H_f \geq H_{\text{quantum}}.$$

### B.4 FINAL SOLUTION

Finally, the solution $H_{\text{HiQ-Lip}}$ obtained after the refinement phase satisfies the following:

$$H_{\text{HiQ-Lip}} \geq H_f \geq H_{\text{quantum}} \geq H_c \geq \min H.$$

This indicates that HiQ-Lip estimates the global Lipschitz constant with an upper bound, and the method is designed to provide a safe approximation, suitable for tasks like robustness certification where an upper bound is preferred.

