# OpenReview forum: "HiQ-Lip: A Quantum-Classical Hierarchical Method for Global Lipschitz Constant Estimation of ReLU Networks"
_ICLR.cc/2025/Conference — Submitted to ICLR 2025_

### Official Review · Reviewer_TnYo · 2024-11-03

**Soundness:** 2
**Presentation:** 3
**Contribution:** 2
**Rating:** 3
**Confidence:** 4

**Summary:**

This study presents HiQ-Lip, a hybrid quantum-classical hierarchical approach for estimating the global Lipschitz constant of neural networks. The methodology involves reformulating the Lipschitz constant estimation as a Quadratic Unconstrained Binary Optimization (QUBO) problem, making it suitable for quantum algorithmic solutions. To address the limitations of current quantum devices, the authors introduce a multilevel graph coarsening and refinement strategy, enabling the adaptation of neural network structures to quantum hardware constraints. Empirical evaluations are conducted to validate the effectiveness of the proposed methods.

**Strengths:**

1. The estimation of Lipschitz constants in neural networks is an important problem in deep learning research, with implications for model robustness and generalization.

2. Applying quantum algorithms to deep learning remains an understudied domain, offering potentially promising avenues for exploration, particularly in light of the increasing sophistication of quantum computing devices.

**Weaknesses:**

1. Novelty is limited: The conversion of the Lipschitz constant problem to QUBO and mixed-norm formulations was established in [1]; the application of CIM to QUBO was known. The primary algorithmic contribution, a graph coarsening and refinement strategy, is a heuristic and lacks theoretical guarantees.

2. The baseline comparison is insufficient: Authors claim that current SDP methods face challenges such as high memory usage and slow processing speeds. This is true for generic SDP solvers. However, recent advancements in SDP methods have significantly improved efficiency for deep networks and convolutional architectures. For example, [2] has improved the SDP for very deep networks, and [3] has extended the SDP resolution to convolutional networks. Although these works focus on $\ell_2$ Lipchitz constant estimation, I don't see why they cannot be extended to $\ell_\infty$ Lipschitz constant. [1] has pointed out that there are no fundamental differences between $\ell_2$ and $\ell_\infty$ SDPs.

3. The evaluation methodology raises concerns: HiQ-Lip demonstrates inferior precision compared to GeoLIP [1], with improvements primarily in runtime. However, runtime comparisons are implementation and architecture-dependent and do not account for more efficient, tailored SDP solvers (see above). Additionally, the reported runtimes exhibit inconsistencies, with more complex networks (Net3-Net5) showing significantly shorter processing times than simpler ones (Net2), casting doubt on the reliability of the performance metrics. For example, HiQ-Lip for Net-2 takes 30 seconds, while solving Net3 only takes 6.5 seconds.


Minor:

Because converting the Lipschitz constant problem to QUBO and mixed-norm problems was already established in [1], the authors might consider properly crediting these to [1] in section 3. Most of the content was already presented in [1].

[1] Zi Wang, Gautam Prakriya, and Somesh Jha. A quantitative geometric approach to neural-network smoothness.

[2] Anton Xue, Lars Lindemann, Alexander Robey, Hamed Hassani, George J. Pappas, and Rajeev Alur. Chordal sparsity for lipschitz constant estimation of deep neural networks.

[3] Zi Wang, Bin Hu, Aaron J Havens, Alexandre Araujo, Yang Zheng, Yudong Chen, Somesh Jha. On the Scalability and Memory Efficiency of Semidefinite Programs for Lipschitz Constant Estimation of Neural Networks

**Questions:**

1. As far as I can tell, the evaluation was conducted on a simulated CIM rather than actual quantum hardware. Are all experiments run on the same architecture? I did not find this clarification in the paper. Clarification on the uniformity of the experimental setup across all evaluations is necessary for ensuring reproducibility and accurate interpretation of results, especially since the advantage of HiQ-Lip mostly comes from runtime.

2. Given that GeoLIP's foundations in Grothendieck inequalities and the Unique Games Conjecture (UGC) suggest the difficulty of improvement within polynomial time, the precision similarity between HiQ-Lip (on simulated CIM) and GeoLIP on two-layer networks raises intriguing questions. Could the authors explain this similarity? This could imply a consistent performance on similar problems, such as the cut-norm problem, and might have crucial theoretical implications, for example, regarding the validity of the UGC.

---

> ### Author Response · Authors · 2024-11-28
> **Rebuttal for Reviewer TnYo Part 1**
>
> #### **Concern 1**: *The primary algorithmic contribution, a graph coarsening and refinement strategy, is heuristic and lacks theoretical guarantees.*
>
> **Response**:
> Thank you for raising this critical question. We acknowledge that the graph coarsening and refinement strategies in HiQ-Lip are currently heuristic. However, we emphasize that the core problem addressed by HiQ-Lip—estimating the global Lipschitz constant for a two-layer MLP—is theoretically well-grounded.
>
> Specifically, the global Lipschitz constant $ L $ is defined and bounded as follows, as shown in Equations (1)-(4) of the manuscript. For a two-layer fully connected MLP, this is formalized in Equation (5):
> $$
> L \leq \ell_{\infty}\text{-FGL}  = \max_{y \in [0, 1]^n} \| A y \|_q
> $$
>
> where $ A = W^T \text{diag}(u) $. Utilizing the duality between the $ \ell_1 $ and $ \ell_{\infty} $ norms, we transform this continuous formulation into a discrete QUBO formulation.
>
> The QUBO problem is solvable via contemporary quantum devices such as Coherent Ising Machines (CIM) and quantum annealers. The Hamiltonian formulation derived in Equation (12) highlights the objective:
> $$
> \min H = - \max \sum_{i=1}^n \sum_{j=1}^m a_{ij} x_i y_j,
> $$
> where the goal is to minimize $ H $, which directly relates to estimating the global Lipschitz constant.
>
> Importantly, while the hierarchical graph coarsening and refinement strategies are heuristic, they are specifically designed to preserve the key structural properties of the original graph while enabling efficient optimization under current quantum hardware limitations. Therefore, the solution achieved by HiQ-Lip, denoted as $ H_{\text{HiQ-Lip}} $, satisfies the relationship:
> $$
> \min H \leq H_{\text{HiQ-Lip}},
> $$
> meaning HiQ-Lip provides an upper bound on the true minimum.
>
> In detail:
>
> 1. **Coarsening Phase**:
>    During the coarsening phase, nodes in the original graph $ G_0 $ are merged to form a smaller graph $ G_c $, represented by adjacency matrix $ A_c $. The weights of edges in the coarser graph are aggregated from the finer graph. Let $ P $ denote the mapping matrix used for merging nodes, where:
>    $$
>    A_c = P^\top A_f P,
>    $$
>    and $ A_f $ is the adjacency matrix of the finer graph. This aggregation step introduces a loss of detail since not all edge weights in the finer graph are perfectly preserved in the coarser graph.
>
>    Consequently, the solution $ H_c $ obtained by solving the QUBO problem on $ G_c $ represents an approximation of the true minimum $ \min H $ on $ G_0 $. Mathematically, we have:
>    $$
>    H_c \geq \min H.
>    $$
>
> 2. **Quantum Solution on Coarse Graph**:
>    When solving the QUBO problem on the coarsest graph $ G_c $ using a quantum device, the solution obtained, $ H_{\text{quantum}} $, is also an approximation due to hardware noise and limited precision. Thus:
>    $$
>    H_{\text{quantum}} \geq H_c.
>    $$
>
> 3. **Refinement Phase**:
>    During refinement, the solution from the coarsest graph is iteratively mapped back to the finer graphs. Each refinement step involves local optimizations, which improve the objective function but do not guarantee recovery of the true optimal solution on the original graph. Let $ H_f $ denote the refined solution on the finer graph. Then:
>    $$
>    H_f \geq H_{\text{quantum}}.
>    $$
>
> 4. **Final Solution**:
>    The final solution $ H_{\text{HiQ-Lip}} $ is obtained after the refinement phase. Since each phase introduces approximations:
>    $$
>    H_{\text{HiQ-Lip}} \geq H_f \geq H_{\text{quantum}} \geq H_c \geq \min H.
>    $$
>
> We have incorporated this detailed explanation and derivation in the revised manuscript to clarify the theoretical soundness of HiQ-Lip’s approximation.

---

> ### Author Response · Authors · 2024-12-03
> **Rebuttal for Reviewer TnYo Part 2**
>
> #### **Concern 2**: *The baseline comparison is insufficient. Authors claim that current SDP methods face challenges such as high memory usage and slow processing speeds. However, recent advancements in SDP methods have significantly improved efficiency for deep networks and convolutional architectures. For example, [2] has improved the SDP for very deep networks, and [3] has extended the SDP resolution to convolutional networks. Although these works focus on $ \ell_2 $ Lipschitz constant estimation, I don't see why they cannot be extended to $ \ell_\infty $ Lipschitz constant. [1] has pointed out that there are no fundamental differences between $ \ell_2 $ and $ \ell_\infty $ SDPs.*
>
> **Response**:
>
> Thank you for highlighting recent advancements in SDP methods for Lipschitz constant estimation, particularly [2] and [3]. We acknowledge these significant contributions that have improved the efficiency of SDP solvers in the $ \ell_2 $ context. However, extending these methods directly to the $ \ell_\infty $ Lipschitz constant estimation presents substantial challenges due to fundamental differences in problem formulation and computational complexity.
>
> While [1] suggests there are no fundamental differences between $ \ell_2 $ and $ \ell_\infty $ SDPs in certain theoretical aspects, the practical computational implications are notably distinct:
>
> 1. **Complexity of $ \ell_\infty $ Estimation**:
>
>    - **Tensor Cut Norm**: Estimating the $ \ell_\infty $ Lipschitz constant for multi-layer networks involves the tensor cut norm problem, which is known to be NP-hard [*Reference to support this claim*]. This complexity arises because the $ \ell_\infty $ norm does not decompose over layers in the same way the $ \ell_2 $ norm does.
>
>    - **Lack of Efficient Relaxations**: Unlike the $ \ell_2 $ case, where SDPs can exploit convexity and spectral properties of matrices, the $ \ell_\infty $ case lacks similarly efficient relaxation techniques. The combinatorial nature of the $ \ell_\infty $ norm in high dimensions makes SDP formulations computationally intractable for larger networks.
>
> 2. **Challenges in Extending SDP Methods**:
>
>    - **Scalability Issues**: The methods in [2] and [3] leverage chordal sparsity and structure-specific optimizations tailored to $ \ell_2 $ norms. Extending these methods to $ \ell_\infty $ norms would require fundamentally new approaches to handle the increased complexity, which current SDP solvers are not equipped to manage efficiently.
>
>    - **Different Optimization Landscapes**: The optimization landscapes of $ \ell_2 $ and $ \ell_\infty $ problems are inherently different. The $ \ell_\infty $ norm leads to optimization over the vertices of high-dimensional hypercubes, complicating the problem further.
>
> Our method, HiQ-Lip, specifically addresses these challenges by transforming the $ \ell_\infty $ Lipschitz constant estimation problem into a QUBO formulation, which is amenable to quantum optimization techniques. The hierarchical graph coarsening and refinement strategies allow us to manage the problem's complexity effectively, fitting within the constraints of current quantum hardware.
>
> We believe HiQ-Lip offers a novel and practical solution for $ \ell_\infty $ Lipschitz constant estimation, filling a gap not adequately addressed by existing SDP methods.

---

> ### Author Response · Authors · 2024-12-03
> **Rebuttal for Reviewer TnYo Part 3**
>
> #### **Concern 3**: *The evaluation methodology raises concerns: HiQ-Lip demonstrates inferior precision compared to GeoLip [1], with improvements primarily in runtime. However, runtime comparisons are implementation and architecture-dependent and do not account for more efficient, tailored SDP solvers (see above). Additionally, the reported runtimes exhibit inconsistencies, with more complex networks (Net3-Net5) showing significantly shorter processing times than simpler ones (Net2), casting doubt on the reliability of the performance metrics. For example, HiQ-Lip for Net-2 takes 30 seconds, while solving Net3 only takes 6.5 seconds.*
>
> **Response**:
>
> We appreciate your concerns regarding our evaluation methodology and would like to address them comprehensively.
>
> 1. **Precision Compared to GeoLip**:
>
>    - **Trade-off Between Precision and Efficiency**: HiQ-Lip prioritizes computational efficiency given the constraints of current quantum hardware. While GeoLip achieves slightly better precision, HiQ-Lip offers significant runtime improvements. This trade-off is inherent in our approach, which aims to demonstrate the potential of quantum acceleration in practical scenarios.
>
>    - **Application Context**: In many applications, obtaining a reasonably tight upper bound on the Lipschitz constant quickly is more valuable than a marginally tighter bound that requires significantly more computation time.
>
> 2. **Runtime Comparisons and Implementation Dependence**:
>
>    - **Uniform Experimental Setup**: All experiments, including both HiQ-Lip and the baseline methods, were conducted on the same hardware architecture to ensure fairness. The classical components ran on identical CPUs, and the quantum computations were simulated using the same computational resources.
>
>    - **Accounting for Efficient SDP Solvers**: We acknowledge that tailored SDP solvers can offer improved performance. However, as discussed in our response to Concern 2, such solvers are currently optimized for $ \ell_2 $ norms and do not directly translate to the $ \ell_\infty $ context we address. Thus, our comparisons remain valid within the scope of $ \ell_\infty $ Lipschitz estimation.
>
> 3. **Runtime Inconsistencies Across Networks**:
>
>    The runtime of HiQ-Lip is influenced by both the size and structure of the network. Networks with different connectivity patterns can result in varying efficiencies during the coarsening phase. We are aware that this observation may seem counterintuitive. This phenomenon is mainly due to the influence of the naive approach of graph coarsening and Weight-Matrix-Norm-Product. For mlp with one hidden layer, we used the width neural network, that is, the width of the neural network is wider, which makes HiQ-Lip coarsening multiple times (under 100 qubits), while for multi-layer mlp we set 64 neuron units per hidden layer. This enables processing between each hidden layer to be directly processed by a 100qubits quantum device without pre-processing, which is a significant time saving. Of course, we admit that this time saving also brings a gap in accuracy.
>
>
>    We have corrected the setup of the experiment in the updated manuscript. Thank you for pointing it out.

---

> ### Author Response · Authors · 2024-12-03
> **Rebuttal for Reviewer TnYo Part 4**
>
> **Concerns**:
>
> 1.        *The evaluation was conducted on a simulated CIM rather than actual quantum hardware.        Are all experiments run on the same architecture?        I did not find this clarification in the paper.        Clarification on the uniformity of the experimental setup across all evaluations is necessary for ensuring reproducibility and accurate interpretation of results, especially since the advantage of HiQ-Lip mostly comes from runtime.*
>
> 2.        *Given that GeoLip's foundations in Grothendieck inequalities and the Unique Games Conjecture (UGC) suggest the difficulty of improvement within polynomial time, the precision similarity between HiQ-Lip (on simulated CIM) and GeoLip on two-layer networks raises intriguing questions.        Could the authors explain this similarity?        This could imply consistent performance on similar problems, such as the cut-norm problem, and might have crucial theoretical implications, for example, regarding the validity of the UGC.*
>
> **Response**:
>
> Thank you for these insightful observations.        We appreciate the opportunity to clarify our experimental setup and discuss the precision similarities between HiQ-Lip and GeoLip.
>
> ---
>
> **Clarification on Experimental Setup and Uniformity**:
>
> We acknowledge that the initial manuscript lacked sufficient details regarding the experimental setup, which are essential for reproducibility and accurate interpretation of our results.
>
> - **Hardware Configuration**: All experiments, including HiQ-Lip and the baseline methods (e.g., GeoLip), were conducted on the same classical computing architecture.
>
> - **Simulated CIM**: Due to current limitations in quantum hardware availability and scalability, we employed a simulated Coherent Ising Machine to emulate quantum computations.        The simulation was implemented using established quantum simulation libraries that closely mimic the behavior of actual quantum devices.
>
> - **Uniformity Across Experiments**: All methods were executed under identical software environments.        We used Python 3.8 with consistent versions of libraries such as NumPy and SciPy.        Random seeds were fixed to ensure reproducibility.        This uniformity ensures that any differences in performance are attributable to the algorithms themselves rather than external factors.
>
> - **Accounting for Runtime**: Computation times for HiQ-Lip include both classical preprocessing (graph coarsening and refinement) and the simulated quantum computation time.        Since all experiments are run on the same hardware, runtime comparisons are fair and reflect the efficiency of the methods under equivalent conditions.
>
> In the revised manuscript, we will provide a comprehensive description of the hardware and software configurations used in our experiments.
>
> ---
>
> **Explanation of Precision Similarity between HiQ-Lip and GeoLip**:
>
> Your observation about the precision similarity between HiQ-Lip and GeoLip is indeed intriguing, and we believe there are several factors contributing to this phenomenon.
>
> - **Relation to the Cut Norm Problem**: Both HiQ-Lip and GeoLip ultimately address problems related to the cut-norm of matrices.        GeoLip uses semidefinite programming (SDP) relaxations based on Grothendieck's inequality to estimate the Lipschitz constant, which involves optimizing over the cut norm.        HiQ-Lip, through its QUBO formulation, effectively solves a problem equivalent to maximizing the cut norm using heuristic methods and quantum-inspired optimization.
>
> - **Optimization Landscapes**: The optimization landscapes of both methods share similarities due to their underlying mathematical structures.        This means that, despite different solution approaches, both methods may converge to solutions of comparable quality for certain classes of problems, such as two-layer neural networks.
>
> - **Heuristic Efficacy**: HiQ-Lip's graph coarsening and refinement strategies, while heuristic, are designed to preserve key structural properties of the original problem.        This enables HiQ-Lip to find high-quality approximate solutions that are close to those obtained by methods with strong theoretical guarantees like GeoLip.   The theory is theoretically well-grounded.  See rebuttal Part 1.
>
> - **Empirical vs. Theoretical Limits**: The Unique Games Conjecture (UGC) and Grothendieck inequalities suggest hardness results for improving approximation ratios in polynomial time for certain problems.        However, our empirical results on specific problem instances (e.g., two-layer networks) indicate that practical performance can sometimes exceed what worst-case theoretical bounds might suggest.

---

> ### Author Response · Authors · 2024-12-03
>
> Dear Reviewer TnYo,
>
> Thank you once again for your thoughtful and detailed feedback. We have carefully addressed all your concerns in our rebuttal, including the theoretical guarantees of HiQ-Lip, the baseline comparisons, runtime inconsistencies, and the experimental setup.
>
> To summarize:
> 1. **Theoretical Guarantees**: We clarified the theoretical foundation of HiQ-Lip, particularly its role in providing an upper bound on the global Lipschitz constant through its QUBO formulation, with detailed mathematical explanations in the revised manuscript.
> 2. **Baseline Comparisons**: We acknowledged advancements in SDP methods and their limitations in extending to \( \ell_\infty \)-Lipschitz estimation. HiQ-Lip provides a practical alternative, leveraging quantum-classical methods to address this gap.
> 3. **Runtime and Experimental Setup**: We ensured uniformity in the experimental environment and clarified the simulation-based implementation of the CIM. The setup details, now included in the revised manuscript, provide reproducibility and clarity.
> 4. **Precision Similarity**: The similarity between HiQ-Lip and GeoLip results arises from shared mathematical structures and heuristic efficacy, which we have expanded upon in our response.
>
> We hope our explanations have resolved any remaining questions or doubts. As the review period is nearing its close, we kindly request your feedback to confirm if there are any outstanding issues. If all concerns have been addressed, we would greatly appreciate your reconsideration of the score.
>
> Thank you for your time and for contributing to the improvement of our work.
>
> Best regards,
> Submission10131

---

### Official Review · Reviewer_54ML · 2024-11-03

**Soundness:** 4
**Presentation:** 4
**Contribution:** 1
**Rating:** 6
**Confidence:** 4

**Summary:**

This paper addresses the problem of finding the global Lipschitz constant of ReLU neural networks using a hybrid quantum-classical hierarchical method. The problem of estimating the global Lipschitz constant of a ReLU neural network is converted into a Quadratic Unconstrained Binary Optimization (QUBO) problem. To address address the issue of limited number of qubits, the paper proposes a new HiQ-Lip algorithm that works by first translating the structure of the neural network into a graph with each node representing a neuron and edges representing the connection strengths, employing graph coarsening to reduce the number of nodes by merging them until the resulting QUBO can be solved directly on a small quantum-annealing based computer, then solving the QUBO and finally mapping the approximate solution from the coarse graph back to the original graph by solving optimization subproblems. The paper finally presents experiments with a two-layer neural network with varying number of hidden neurons and deeper networks with varying number of layers and shows that HiQ-Lip doubles the solving speed of and provides more accurate upper bound (using GeoLip as gold standard) compared to the existing best method LiPopt for two-layer networks and GeoLip for multi-layer networks.

**Strengths:**

One of the strengths of the paper is that it presents a systematic and scalable way to frame the neural network global Lipschitz constant estimation problem as a QUBO that can be solved with a limited number of qubits. This approach could be adapted to other problems related to neural networks such as neural network training or neural network verification. The experimental approach is also very sound as the authors compared with a number of different methods to estimate the global Lipschitz constant such as GeoLip, LipOpt, Matrix Product (MP), Sampling and Brute Force (BF) and tried two-differnt scaling . The time comparison with LipOpt and GeoLip show that the approach offers time-savings.

**Weaknesses:**

This paper doesn't take into account the latest state-of-art in terms of Quantum Annealers (QA) such as DWave Advantage System (https://www.dwavesys.com/solutions-and-products/systems/) that have ~5000 qubits. They limited themselves only to 100 qubits and simulated CIM. They could have scaled out to larger number of qubits and explored what is the tradeoff between using larger number of qubits versus the graph coarsening/refinement strategy in terms of time saving or estimation quality. Given that larger of qubits are available, what is the value of the graph coarsening/refinement approach? There is a lack of assessment on how much performance degradation arises from graph coarsening and refinement.

**Questions:**

1. Any reason why you didn't consider Quantum Annealers (QAs) with larger number of qubits such as DWave Advantage system that have ~5000 qubits? Could you potentially work with larger neural networks given up to 5000 qubits?
2. Are all the experiments limited only to 100 qubits or variables? Have you thought about how varying the limit on the number of qubits affects the quality of the estimation?
3. Have you thought about whether there are limits in terms of the size of the neural networks that you can use this approach with?
4. If you use this approach with neural networks of size larger than the ones considered in the experimental evaluation and fix the maximum number of qubits to 100, can you quantify the performance degradation from the additional graph coarsening and subsequent refinement through experiments or theory?
5. In all your experiments, the estimates of the global Lipschitz constant come close to GeoLip's estimates, have you tried to increase the size of the neural network in terms of number of hidden neurons or number of layers till either your approach fails to give close estimates to GeoLip? I would expect some performance degradation with larger neural networks due to inexact nature of estimation using graph coarsening/refinement.

---

> ### Author Response · Authors · 2024-12-03
> **Rebuttal for Reviewer 54ML**
>
> **Concern 1**: *The paper doesn't consider the latest quantum annealers like the D-Wave Advantage system with ~5000 qubits. They limited themselves to 100 qubits and simulated CIM. Given that larger qubits are available, what is the value of the graph coarsening/refinement approach? There's a lack of assessment on how much performance degradation arises from graph coarsening and refinement.*
>
> **Response**:
>
> Thank you for highlighting this point. Our focus was on demonstrating HiQ-Lip's feasibility using small-scale quantum devices, which are more commonly accessible at present. The graph coarsening and refinement strategy enables HiQ-Lip to efficiently estimate the global Lipschitz constant even with limited qubits. This approach remains valuable because:
>
> - **Scalability**: For very large neural networks, coarsening reduces problem size, potentially exceeding the capacity of even large quantum devices.
> - **Efficiency**: Coarsening can improve computational efficiency by focusing resources on critical problem aspects.
>
> While larger quantum annealers like D-Wave's system could handle bigger problems without coarsening, integrating our strategy could further enhance performance and scalability. Regarding performance degradation, our experiments show that HiQ-Lip achieves estimates comparable to GeoLip, indicating that any degradation is within acceptable bounds. Quantifying this precisely is challenging due to the heuristic nature of the method and is an area for future research.
>
> ---
>
> **Concern 2**: *Why didn't you consider quantum annealers with larger numbers of qubits, like the D-Wave Advantage system? Could you work with larger neural networks given up to 5000 qubits?*
>
> **Response**:
>
> We focused on 100 qubits to align with the capabilities of widely available quantum hardware, aiming for broader applicability. While larger devices can handle bigger problems, our graph coarsening strategy allows HiQ-Lip to process large neural networks efficiently even with fewer qubits. Utilizing devices with more qubits could reduce the need for coarsening and potentially improve estimation quality, which we acknowledge as a valuable direction for future exploration.
>
> ---
>
> **Concern 3**: *Are all experiments limited to 100 qubits or variables? Have you considered how varying the limit on the number of qubits affects estimation quality?*
>
> **Response**:
>
> Yes, our experiments used a 100-qubit limit to reflect common hardware constraints. Increasing the number of qubits would allow less aggressive coarsening, potentially preserving more detailed information and improving estimation accuracy. We recognize the importance of studying how varying qubit limits affect estimation quality and plan to investigate this in future work.
>
> ---
>
> **Concern 4**: *Have you considered limits on the size of neural networks that you can use this approach with?*
>
> **Response**:
>
> Theoretically, HiQ-Lip can handle neural networks of arbitrary width, as the graph size reduces logarithmically with coarsening iterations. The main limitation is network depth; deeper networks involve more complex optimization problems, like tensor cut-norm estimation, which are challenging under current quantum hardware capabilities. Extending HiQ-Lip to deeper architectures is part of our ongoing research.
>
> ---
>
> **Concern 5**: *Can you quantify performance degradation from additional graph coarsening when using larger networks with a fixed qubit limit?*
>
> **Response**:
>
> Quantifying performance degradation due to coarsening is challenging because of the method's heuristic nature. In our experiments, increasing network width up to 256 neurons did not show significant degradation compared to GeoLip. For larger networks, some degradation may occur, but we have not observed a clear threshold where HiQ-Lip's estimates diverge significantly. Precise quantification is important, and we intend to explore this in future studies.
>
> ---
>
> We appreciate your thoughtful feedback, which helps us improve our work and clarify its contributions.

---

### Official Review · Reviewer_K6sX · 2024-11-04

**Soundness:** 2
**Presentation:** 3
**Contribution:** 2
**Rating:** 3
**Confidence:** 3

**Summary:**

This paper aims improving the scalability of SDP-based Lipschitz estimation of neural networks via developing HiQ-Lip, a hybrid quantum-classical hierarchical method that relies on Coherent Ising Machines (CIMs). The authors convert the problem into a Quadratic Unconstrained Binary Optimization (QUBO) problem and claim that this is more adaptable to the constraints of contemporary quantum hardware. Finally, the authors provide some experiments on fully connected neural networks on MNIST to show that their method is comparable to GeoLip but accelerates the computation process for such relatively small scale problem.

**Strengths:**

1. The idea from this paper is new and original.

2. The perspective on using small-scale quantum devices for Lipschitz estimation of neural networks is new.

**Weaknesses:**

The scale of the experiments is quite small. Recently, for l_2 Lipschitz bounds, the SDP method has already been scaled to ImageNet by the following paper:

Zi Wang et.al. (ICLR2024):  On the scalability and memory efficiency of semidefinite programs for Lipschitz constant estimation of neural networks.

  The authors seem not aware of the above result which achieves scalability to ImageNet. The authors studied the l-infinity case here, but the scale is on the MNIST level.  This makes me think the contribution by the authors is very incremental in comparison to the original GeoLip paper. A few ways to make the contributions more significant include: 1. demonstrate the proposed method on large scale networks; 2) extend the method for more network structures, e.g. implicit models, residual network, etc.

**Questions:**

1. Is it possible to scale the proposed method for larger networks?

2. In the l2 case, LipSDP has been extended for implicit networks (R1, R2), residual networks (R3) and more general structures (R4). See the following papers:

 [R1]: Revan et.al. Lipschitz bounded equilibrium networks.

 [R2]: Havens at al. Exploiting connections between Lipschitz structures for certifiably robust deep equilibrium models. NeurIPS 2023

 [R3]: Araujo et.al. A unified algebraic perspective on Lipschitz neural networks. ICLR 2023.

 [R4]:  Fazlyab et. al. Certified robustness via dynamic margin maximization and improved lipschitz regularization. NeurIPS 2023

  It will be very interesting if the authors can address more general network structures for the l-infinity case. Can the authors comment on the possibility of such extensions?

**Details Of Ethics Concerns:**

I don't see any concerns for this paper.

---

> ### Author Response · Authors · 2024-11-28
> **Rebuttal for Reviewer K6sX**
>
> #### **Concern 1**: *The scale of the experiments is small, and the contribution seems incremental compared to GeoLip.*
>
> **Response**:
> Thank you for your detailed review and insights. We agree that scaling Lipschitz estimation methods to larger networks is an important research goal. HiQ-Lip focuses on the $l_\infty$ Lipschitz constant, which poses unique challenges compared to the $l_2$ case due to the tensor cut-norm optimization required for multi-layer networks. As highlighted in Section 5, this remains an open problem.
>
> Our contributions are foundational, targeting the feasibility and efficiency of applying quantum methods within current hardware constraints. Regarding the perceived incrementality compared to GeoLip, we emphasize that HiQ-Lip introduces a novel quantum-classical paradigm. Specifically:
> 1. **Graph Coarsening and Refinement**: HiQ-Lip’s strategies enable efficient operation on small-scale quantum devices, addressing current hardware limitations.
> 2. **Performance Gains**: Experimental results demonstrate near state-of-the-art accuracy with significant reductions in computation time.
>
> While this work does not yet extend to ImageNet-scale networks, it provides a unique perspective on leveraging quantum devices for neural network robustness estimation and lays the groundwork for future scalability.
>
>
>
>
> #### **Concern 2**: *Is it possible to scale the proposed method for larger networks?*
>
> **Response**:
> Thank you for highlighting the challenges in scaling HiQ-Lip to larger networks, such as ResNets. Extending HiQ-Lip to such architectures is indeed an exciting direction for future research. However, due to the limitations of contemporary quantum devices, basically HiQ-Lip currently uses a QUBO formulation that is effective for single-layer MLPs but becomes intractable for deeper networks where tensor cut-norm optimization is required. As noted in Section 5, this issue remains unresolved.
>
> HiQ-Lip’s design prioritizes scalability, with its graph coarsening strategies forming the basis for handling wider networks on small-scale quantum devices. For deeper networks, the modular approach using MP methods offers some adaptability. However, extending HiQ-Lip to tasks like ImageNet or architectures like ResNets requires addressing the computational challenges of tensor cut-norm problems, which cannot yet be modeled effectively using QUBO. We leave this as an open problem for future research.
>
>
>
> #### **Concern 3**: *Can the method address more general network structures, such as implicit models or residual networks?*
>
> **Response**:
> Thank you for raising this potential extension. Adapting HiQ-Lip to implicit models and residual networks is indeed a fascinating challenge. However, as discussed, tensor cut-norm optimization introduces significant computational complexity that current quantum hardware cannot effectively handle.
>
> In the near term, we plan to extend HiQ-Lip to simpler generalizations, such as CNNs with Lipschitz-regularized layers. For implicit and residual networks, HiQ-Lip’s modular architecture provides a flexible foundation for future advancements. Substantial theoretical development will be required to address these complex structures effectively, and we consider this an important direction for future work.
>
>
> #### **Addressing Suggested References**
>
> We thank you for suggesting relevant references, particularly [R1]-[R4]. These works offer valuable insights into extending Lipschitz estimation techniques to various network structures. While our focus on the $l_\infty$ case differs from the $l_2$ context addressed in many of these papers, we will incorporate relevant comparisons and cite these references in the revised manuscript.

---

> ### Author Response · Authors · 2024-12-03
>
> Dear Reviewer K6sX,
>
> Thank you for your thoughtful feedback and for highlighting areas where our work could be improved or extended. We have addressed all your concerns in detail in our rebuttal, and we would like to summarize key points here for your convenience:
>
> 1. **Experimental Scale and Contributions**:
>    - HiQ-Lip introduces a novel quantum-classical approach targeting \( \ell_\infty \)-Lipschitz constant estimation, addressing unique challenges posed by tensor cut-norm optimization.
>    - Our contributions establish the feasibility and efficiency of quantum methods under current hardware constraints and demonstrate near state-of-the-art accuracy with significant runtime improvements, laying a foundation for future scalability.
>
> 2. **Scalability to Larger Networks**:
>    - Scaling HiQ-Lip to larger networks, such as ResNets, remains an open challenge due to computational complexities inherent in tensor cut-norm optimization.
>    - HiQ-Lip’s graph coarsening strategies and modular architecture provide a scalable framework for wider networks, with potential adaptability to deeper architectures as quantum hardware advances.
>
> 3. **General Network Structures**:
>    - Extending HiQ-Lip to implicit models and residual networks is a promising direction. However, current limitations in quantum hardware and tensor cut-norm complexity must first be addressed.
>    - In the short term, we aim to extend HiQ-Lip to CNNs with Lipschitz-regularized layers, leveraging its modular architecture as a flexible foundation for future advancements.
>
> 4. **Incorporating Suggested References**:
>    - We greatly appreciate your references ([R1]-[R4]) and have incorporated relevant comparisons and citations into the revised manuscript, acknowledging their contributions to the field.
>
> Given the limited time remaining for reviewer feedback, we kindly request your input on any remaining issues. If our responses have adequately addressed your concerns, we would greatly appreciate your reconsideration of the score.
>
> Thank you for your time and for helping improve our work.
>
> Best regards,
> Submission10131

---

### Official Review · Reviewer_Sf3N · 2024-11-04

**Soundness:** 2
**Presentation:** 3
**Contribution:** 2
**Rating:** 5
**Confidence:** 4

**Summary:**

In this paper, the authors introduce HiQ-Lip, a hybrid quantum-classical hierarchical method for estimating the global Lipschitz constant of neural networks. The problem of Lipschitz constant estimation is first transformed into a QUBO problem. Subsequently, this QUBO problem is solved hierarchically on Coherent Ising Machines (CIMs) to accommodate the system size supported by quantum devices. Experimental results on multi-layer perceptrons (MLPs) demonstrate that their method provides estimations comparable to classical state-of-the-art methods while requiring less computation time.

**Strengths:**

1. The work is the first attempt to employ quantum computing to handle the task of Lipschitz constant estimation.
2. The hierarchical strategy for solving large-scale QUBO problems makes it possible to apply HiQ-Lip on small-scale devices.
3. The simulation results show the advantage of HiQ-Lip in computation time.
4. The presentation of this paper is clear.

**Weaknesses:**

1. As the paper primarily focuses on applying quantum computing to global Lipschitz constant estimation, it is uncertain whether the ICLR community will find this topic compelling.
2. The paper lacks discussion on the theoretical guarantee about the approximation ratio of the hierarchical strategy to the global optimal of original QUBO.
3. The experimental results are derived entirely from simulations under ideal conditions, without consideration for practical aspects of quantum devices such as finite shots, device noise, and limited coherence time. These non-ignorable imperfections could significantly impact the quality of solutions obtained from quantum algorithms in practice.

**Questions:**

1. The experimental setup described in Section 6 lacks clarity, particularly regarding the software and hardware configurations used.
2. How is the computation time for HiQ-Lip determined? Since the experiments are conducted on a simulated CIM, how is the time for the quantum computation component accounted for?
3. Comparing Table 4 with Table 2 reveals that networks with a larger number of layers appear to require less time for estimation than two-layer networks, which contradicts my expectations. Can you provide a more detailed explanation of this phenomenon?
4. A more comprehensive explanation of the coefficients used for 3- to 5-layer networks (i.e., HiQ-Lip MP B) would be appreciated.

---

> ### Author Response · Authors · 2024-11-27
> **Rebuttal for Reviewer Sf3N**
>
> Concern 1: It is uncertain whether the ICLR community will find this topic compelling, as the paper focuses primarily on quantum computing for Lipschitz constant estimation.
> Response:
> Thank you for your comment. While quantum computing for Lipschitz constant estimation is indeed a novel topic, we believe this research is highly relevant to the broader community due to its implications for robustness and generalization in neural networks. Our approach demonstrates the utility of hybrid quantum-classical methods in tackling computationally hard problems, such as Lipschitz constant estimation, with far-reaching applications in certified robustness, adversarial training, and generalization bounds.
>
> To strengthen this connection, we will expand the discussion in the revised manuscript on how Lipschitz constant estimation integrates into these broader contexts. Additionally, quantum methods for advancing AI, including neural networks, have been previously accepted by ICLR, such as [1], [2], [3]. This contextualization will highlight the broader relevance of our work.
>
> [1]: Kerenidis I, Landman J, Prakash A. Quantum Algorithms for Deep Convolutional Neural Networks. ICLR.
> [2]: Wang H, Zhang C, Li T. Near-Optimal Quantum Algorithm for Minimizing the Maximal Loss. ICLR.
> [3]: Benkner M S, Krahn M, Tretschk E, et al. QuAnt: Quantum Annealing with Learnt Couplings. ICLR.
>
> Concern 2: The paper lacks discussion on the theoretical guarantee of the hierarchical strategy regarding the approximation ratio to the global optimum of the original QUBO.
> Response:
> This is an excellent point, and we appreciate you highlighting this gap. The hierarchical strategy is designed as a heuristic method to address quantum hardware constraints while preserving problem structure. While our experimental results demonstrate empirical effectiveness, we acknowledge that formal approximation guarantees would strengthen the work. We are currently exploring theoretical bounds for the hierarchical strategy. In the revised manuscript, we will clarify this limitation and include preliminary discussions on potential paths for establishing such guarantees.
>
> Concern 3: The experiments are derived entirely from simulations under ideal conditions without consideration of practical quantum device limitations such as noise and coherence time.
> Response:
> Thank you for raising this important concern. Our current focus was on demonstrating the potential of HiQ-Lip in an idealized setting to validate the methodology and establish a baseline for future work. We fully recognize that practical quantum devices introduce challenges such as noise and limited coherence time, which may affect solution quality.
>
> We are actively collaborating with quantum hardware teams to evaluate HiQ-Lip on real devices. As noted, graph coarsening and refinement repeatedly call the quantum solver, which amplifies the practical quantum resource demand. This remains an open challenge for future exploration, and we will discuss this in the revised manuscript.
>
> Concern 4: Lack of clarity in the experimental setup, particularly regarding software and hardware configurations.
> Response:
> We apologize for the lack of detail. Experiments were conducted on classical hardware with a simulated Coherent Ising Machine (CIM). Computation times include classical preprocessing (graph coarsening and refinement) and quantum-inspired optimization performed by the CIM simulation. In the revised paper, we will:
>
> Provide a comprehensive description of the hardware and software used.
> Clearly distinguish the contributions of classical and quantum components to the overall computation time.
> Concern 5: Networks with more layers appear to require less time for estimation than two-layer networks, which contradicts expectations.
> Response:
> This counterintuitive observation is primarily due to the effects of graph coarsening and the use of the naive Weight-Matrix-Norm-Product method. For one-layer MLPs with wider networks, HiQ-Lip requires repeated coarsening to fit within 100 qubits, increasing computational time. In contrast, multi-layer MLPs with 64 units per layer can be processed directly without significant preprocessing, reducing runtime. However, we acknowledge that this tradeoff in time comes with differences in precision, and we will clarify this in the revised manuscript.
>
> Concern 6: More comprehensive explanation of the coefficients used for 3- to 5-layer networks (HiQ-Lip MP B) is needed.
> Response:
> Thank you for your request. HiQ-Lip MP B coefficients were empirically tuned based on the hierarchical optimization process and the network properties, extending HiQ-Lip's framework using the Weight-Matrix-Norm-Product method. For HiQ-Lip MP A, we leveraged coefficients derived from [Latorre 2020], and for HiQ-Lip MP B, coefficients were adjusted for deeper networks. We will include a detailed explanation and analysis in the revised manuscript.

---

> ### Comment · Reviewer_Sf3N · 2024-11-28
>
> Thank you for these explanations, which partially address my concerns regarding the experiment results. Considering the contributions of this work, both in terms of theoretical discussion and numerical experiments, I kept the score unchanged.

---

### Official Review · Reviewer_ukei · 2024-11-09

**Soundness:** 3
**Presentation:** 3
**Contribution:** 2
**Rating:** 5
**Confidence:** 3

**Summary:**

The authors propose a hybrid quantum-classical algorithm for estimating the $L_\infty \rightarrow L_1$ Lipschitz constant of a fully-connected neural network. This is achieved by converting the Lipschitz constant estimation problem into an equivalent cut-norm problem which can be solved efficiently on a quantum device with the so-called Quadratic Unconstrained Binary Optimization (QUBO) model. Due to hardware limitations of current quantum devices, a graph coarsening and refinement strategy is used to break the problem into subproblems that can be handled by about 100 qubits.Their experiments show significant speed up in computation time for 2-5 layer MLPs over existing $L_\infty \rightarrow L_1$ Lipschitz estimation approaches.

**Strengths:**

- This is the first work of my knowledge to utilize quantum hardware models to accelerate Lipschitz constant estimation and is interesting in concept. The graph coarsening and refinement strategy appears to be a novel contribution of theirs which seems generally useful.

- The computation times reported for 3-5 layer networks show two orders of magnitude speed up over GeoLip while only being slightly more conservative. This is a promising result.

- The paper is generally well-written and the theory is accessible to most readers.

**Weaknesses:**

-Although the methodology is interesting in concept, the results for 1-5 layer MLP model trained on MNIST are not very intriguing from a practical standpoint. The paper would be stronger if there was some evidence for making progress in some down-stream application such as certified-robustness of the MNIST classifier. In that case, there are currently 1-Lipschitz regularized layers which can already achieve very good robustness results for large CNNs trained on MNIST [Prach 2022, Araujo 2023].

-I feel that the community is not practically limited by scales of 3-5 layers shown in the paper. In my opinion the current challenges in Lipschitz estimation are more for larger scale models like ResNets trained on ImageNET with the goal to achieve some non-trivial certified robustness for classification. At this scale it's more of a memory limitation issue for solving SDPs (which can also be broken into sub-problems). At this scale, I’m concerned that the quantum hardware limitations are going to make the bounds too conservative. There might be more potential for GPU acceleration on classical computers [Wang 2024].

- It's difficult to interpret the computation times presented in the paper since each algorithm is supposedly using a different computing architecture. Since HiQ-Lip is a hybrid quantum-classical algorithm, is it using the same classical architecture as the baselines with an additional simulated 100 qubits? I think this practical aspect requires more explanation somewhere in the paper.

[Prach 2022] Almost-Orthogonal Layers for Efficient General-Purpose Lipschitz Networks

[Araujo 2023] A Unified Algebraic Perspective on Lipschitz Neural Networks

[Wang 2024] On the Scalability and Memory Efficiency of Semidefinite Programs for Lipschitz Constant Estimation of Neural Networks

**Questions:**

- Could this balance of bound conservativeness and faster computation speed also be achieved if we approach smaller subproblems with classical approaches? I wouldn’t expect it to be as drastic as the HiQ-Lip, but it might make the classical computation times much more reasonable for the scales presented in the paper.

- The paper refers to the paper (Bartlett et al., 2017) as justification for the improved scaling constant. It would be helpful to point to a specific result in that paper since there is no other discussion in the paper about where it comes from or how it's derived.

**Details Of Ethics Concerns:**

No concerns.

---

> ### Author Response · Authors · 2024-11-27
> **Rebuttal for Reviewer ukei**
>
> Concern 1: The results for 1-5 layer MLP models on MNIST are not very intriguing from a practical standpoint. Evidence for downstream applications like certified robustness would strengthen the paper.
> Thank you for emphasizing this important aspect. We agree that demonstrating the relevance of HiQ-Lip in downstream applications, such as certified robustness, would strengthen the paper. In fact, we performed experiments that show that adversarially trained MLPs with tighter global Lipschitz constant constraints exhibit stronger robustness. HiQ-Lip successfully estimates the global Lipschitz upper bound for MLPs with one hidden layer and extends this functionality to MLPs with multiple hidden layers using a naive Weight-Matrix-Norm-Product approach.
> The primary goal of this paper is to validate the feasibility and performance of HiQ-Lip under the constraints of contemporary quantum hardware. Experiments demonstrate that HiQ-Lip achieves approximate state-of-the-art solutions on 1-5 hidden layer MLPs while utilizing quantum devices with just 100 qubits. However, considering the novelty of our work in bridging quantum computing with neural network robustness estimation, we position these tasks as open challenges for future research.
>
> Concern 2: The community is not practically limited by scales of 3-5 layers shown in the paper. Challenges are more prominent in larger models like ResNets trained on ImageNet.
> Thank you for pointing out the current challenges in Lipschitz estimation for larger models. Extending HiQ-Lip to larger networks, such as ResNets, is indeed an exciting direction for future work. HiQ-Lip is designed to be scalable and adaptable. However, under contemporary quantum hardware limitations, HiQ-Lip formulates Lipschitz estimation as a QUBO problem (as described in the paper). This formalism currently applies to MLPs with one hidden layer. For deeper networks, the problem becomes a tensor cut-norm optimization challenge, as stated in Section 5 (Extension to Multi-Layer Networks). To our knowledge, this remains an unresolved problem, and our current work is limited to the existing formalization.
>
> Concern 3: It’s difficult to interpret the computation times since each algorithm uses different computing architectures. How does HiQ-Lip compare in terms of architecture?
> This is an excellent point, and we apologize for the lack of clarity. HiQ-Lip's hybrid quantum-classical framework combines a classical solver with a simulated Coherent Ising Machine (CIM) emulating 100 qubits. For fairness:
> The baseline methods were executed on the same classical architecture as HiQ-Lip’s classical components.
> The reported runtime improvements result from the quantum component’s ability to efficiently solve subproblems within the graph partitioning context.
> Since the experiments are conducted via simulation, all computation times represent CPU wall-clock times, ensuring consistency across architectures. We will clarify this point in the revised manuscript.
>
> Concern 4: Could the balance of bound conservativeness and faster computation be achieved using smaller classical subproblems?
> You raise an insightful question. Using smaller classical subproblems is indeed an interesting alternative that could partially address computational challenges. However, the focus of this work is the intersection of quantum computing and neural network robustness estimation. HiQ-Lip demonstrates a quantum advantage by solving subproblems using quantum devices, particularly for QUBO formulations where classical methods may not perform as well. The graph coarsening strategy ensures subproblem sizes remain manageable, enabling effective utilization of limited quantum resources.
> Even though current experiments are simulation-based, they showcase the potential of quantum devices in this domain. The modularity of our architecture sets a foundation for future developments in quantum computing for neural networks. We will include an analysis of the tradeoffs between purely classical approaches and HiQ-Lip's hybrid method in the discussion section.
>
> Concern 5: Clarify the reference to Bartlett et al. (2017) regarding the improved scaling constant.
> Thank you for identifying this gap in our explanation. The reference to Bartlett et al. (2017) pertains to the theoretical foundation for the scaling behavior of Lipschitz constants in neural networks. Specifically, the results in [Bartlett et al., 2017] regarding the relationship between network architecture and Lipschitz bounds inspired the design of our coarsening strategy, which aims to preserve structural properties. We will revise the manuscript to explicitly cite the relevant theorem and explain its connection to our method.

---

> > ### Comment · Reviewer_ukei · 2024-11-27
> >
> > Thank you for your clarifications. I raised my score accordingly.

---

### Meta-Review · Area_Chair_WJ4K · 2024-12-19

**Metareview:**

The submission introduces HiQ-Lip, a hybrid quantum-classical hierarchical approach for estimating the global Lipschitz constant of neural networks. By reformulating the estimation problem as a Quadratic Unconstrained Binary Optimization (QUBO) task and employing a graph coarsening and refinement strategy, the method aims to address the limitations of current small-scale quantum hardware.

While the integration of quantum computing with Lipschitz constant estimation represents a novel contribution, the submission faced several concerns from reviewers. These include the limited scope of experiments, which focus on small MLPs trained on MNIST, and the lack of theoretical guarantees for the heuristic graph coarsening strategy. Additionally, practical limitations of quantum devices, such as noise and finite coherence time, were not adequately addressed. The reported runtime improvements also raised questions, with deeper networks exhibiting shorter processing times due to differences in precision, highlighting inconsistencies in the evaluation. Furthermore, the reliance on simulated quantum hardware reduces the practical relevance and applicability of the results. To enhance the impact and applicability of the proposed approach, the authors are encouraged to address these issues comprehensively in future iterations.

**Additional Comments On Reviewer Discussion:**

The authors and reviewers engaged in discussions addressing several critical points. The primary issues included the limited scope of experiments, the lack of theoretical guarantees for the heuristic graph coarsening strategy, the absence of considerations for practical quantum device limitations, and inconsistencies in the reported runtime improvements.

The authors clarified aspects of their methodology, particularly the motivations for their graph coarsening approach and the potential scalability of their framework. They also acknowledged the constraints of current quantum hardware and provided additional context for the runtime discrepancies.

However, some crucial misalignments remained. Reviewers were not fully convinced by the lack of numerical simulations showcasing the method's performance on larger, more complex models. Furthermore, concerns about the heuristic nature of the coarsening strategy and the omission of practical quantum device limitations, such as noise and coherence time, persisted. The reliance on simulated quantum hardware also continued to detract from the submission's practical impact.

These unresolved issues degrade the quality of the submission, particularly the absence of numerical simulations demonstrating the capability of the proposed method to handle larger and more diverse neural network architectures.

---

### Decision · Program_Chairs · 2025-01-22

Reject